# "Knowledge regarding cardiopulmonary resuscitation among health assistants in Nepal: A cross-sectional study"

**Bivek Singh**[1,2]*, **Pathiyil Ravi Shankar**[3]

**1** Department of Medicine, National Cardiac Centre, Bashundhara, Kathmandu, Nepal, **2** Resident, Department of Neurology, Shanghai East Hospital, Tongji University School of Medicine, Shanghai, China, **3** IMU Centre for Education, International Medical University, Kuala Lumpur, Malaysia

\* biveksingh123@gmail.com

## Abstract

### Background

Health assistants play a crucial role in healthcare delivery, particularly in remote and rural areas of Nepal. They should have adequate lifesaving and resuscitation skills. Therefore, assessing their cardiopulmonary resuscitation (CPR) knowledge is essential.

### Objective

To evaluate the knowledge of CPR among health assistants (HAs) in Nepal and explore if there were variations in knowledge scores based on the demographic characteristics of the participants.

### Methods

A quantitative cross-sectional research design was used. The study population included HAs registered with the Nepal Health Professional Council (NHPC) who completed three years of training. Non-probability convenience sampling was employed. Data was collected using an online survey based on the 2020 American Heart Association guidelines. Demographic information and participants' knowledge levels were noted.

### Results

The study involved 500 HAs, with the majority being male and working in government hospitals. Most participants were from Madhesh Province, and the median age was 26 years. Only a fraction of the participants had received training in CPR, and none of them had ever performed CPR. The median knowledge scores were higher among males and among respondents from Madhesh, Lumbini, Karnali, and Sudhurpaschim provinces. The HA's knowledge of the correct depth of CPR compression for children (21%) and infants (17.4%) was limited. CPR scores were different according to variables like training, theory understanding, and practice duration, among others. The findings highlighted the need for more

**Data Availability Statement:** Singh, Bivek; Shankar, P Ravi (2023). Knowledge about cardiopulmonary resuscitation among health assistants in Nepal. figshare. Dataset. https://doi.org/10.6084/m9.figshare.23354156.v1.

**Funding:** The author(s) received no specific funding for this work.

**Competing interests:** I have read the journal's policy and the authors of this manuscript have the following competing interests. [P Ravi Shankar is an academic editor of PLOS ONE] This does not alter our adherence to PLOS ONE policies on sharing data and materials.

**Abbreviations:** ACLS, Advanced Cardiac Life Support; AED, Automated External Defibrillator; BLS, Basic Life Support; CPR, Cardiopulmonary Resuscitation; CTEVT, Council of Technical Education and Vocational Training; IOM, Institute of Medicine; IQR, Interquartile Range; NHPC, Nepal Health Professional Council; PCL, Proficiency Certificate Level.

practical training and regular refresher courses to enhance HAs ability to provide life-saving interventions.

## Conclusion

The study revealed less CPR knowledge and a lack of practical training among HAs in Nepal. To improve healthcare outcomes, providing practical training and ongoing education on CPR is crucial. The findings can contribute to curriculum development and policy changes in healthcare delivery.

## Introduction

Nepal, situated between India and China, features diverse regions: the Himalayas, Hills, and Terai (plain lands) [1–3]. This geographical diversity significantly affects lifestyle and healthcare access, evident through geographical challenges and rural physician shortages. Healthcare professionals are concentrated in urban areas, particularly the capital city, Kathmandu [1–3]. In providing healthcare and addressing healthcare disparities, particularly in remote regions, paramedics such as health assistants (HAs) have a crucial role [4]. Modern medical education in Nepal can be traced back to the inception of the Institute of Medicine (IOM) in 1972. Presently, Nepal has a total of 22 medical colleges, showcasing significant growth in the field of medical training within the country [4]. Additionally, the Council of Technical Education and Vocational Training (CTEVT) oversees 47 institutions offering Proficiency Certificate Levels (PCL) in Health Science (General Medicine) [5, 6].

Despite the increased number of medical and paramedical institutions, remote regions of Nepal still face a shortage of physicians and skilled healthcare workers. The training of HAs begun in 1956 at Bir Hospital in Kathmandu and later transitioned to the IOM in 1972 [5]. In 1989, the CTEVT was established as a national autonomous apex body [7]. Currently, there are 70 institutions across the seven provinces of Nepal offering the PCL in Health Science (General Medicine) program, with a total capacity of 2812 students during the 2020/2021 academic year [8].

The program spans three academic years, with the first year focusing on basic science and foundation subjects, the second year covering theory and practical aspects of basic medical science, and the third year emphasizing the practical application of acquired skills and knowledge in health posts or hospitals [9]. Upon completion of the course, graduates are awarded a "PCL in General Medicine" and registered with the NHPC (currently 17,363 HAs are registered) [9, 10]

HAs are recognized as primary healthcare professionals delivering basic services in rural areas. They are also frequently recruited by private hospitals, where they aid doctors. They can also enroll in the Anesthesia Assistant Course to become non-doctor Anesthesia Assistants.

With a population of 29,192,480, Nepal's population is distributed as 53.66% in Terai, 40.25% in Hilly areas, and 6.09% in Himalayan regions [11]. Despite their crucial role, there has been limited research on HAs training quality, specifically in life-saving skills like Basic Life Support (BLS), Advanced Cardiac Life Support (ACLS), and cardiopulmonary resuscitation (CPR). This can affect survival rates and post cardiac arrest outcomes.

The research aims to evaluate CPR knowledge among HAs in Nepal, and study differences, if any, among different subgroups of respondents.

## Materials and methods

A quantitative cross-sectional research design was used to assess the knowledge regarding CPR among HAs in Nepal. Non-probability convenience sampling was employed to select participants for the study.

### Study variables

**Independent variables.** • Years of experience

• Income level

• Type of health center

• Age

• Gender

• Province

**Dependent variable.** • Level of knowledge (regarding cardiopulmonary resuscitation)

Data were collected using an online survey from February 27, 2023, to April 2, 2023. We conducted a comprehensive literature review to gain insights into the existing research related to CPR knowledge assessment. Based on this review, we defined our research objectives, focusing on assessing different aspects of CPR knowledge. Using the literature review and objectives as guidance, we generated a pool of potential questions. The online questionnaire included a demographic profile section, yes/no questions to assess knowledge on CPR and basic life support, and statements with a three-point scale specific to CPR based on the 2020 guidelines provided by the American Heart Association [12]. The questionnaire was first developed in English and content validation was done utilizing the services of experts in CPR and survey design. The experts provided valuable feedback on question relevance, clarity, and appropriateness. They also suggested modifications to improve the questionnaire's quality. The original questionnaire and the suggested modifications are shown in the Appendix. The consent form and questionnaire were translated into Nepali and then back translated into English by a different set of persons fluent in both languages. The two versions were compared for discrepancies. Both the English and Nepali versions were made available to the participants, and they were free to respond in either language. In Nepal, the medium of instruction for the HA course is English.

The population size was 17,363, representing the total number of registered health assistants with the Nepal Health Professional Council [13]. To determine the sample size, we utilized the online Raosoft sample size calculator (http://www.raosoft.com/samplesize.html), considering a 95% confidence interval and a 5% margin of error. Based on this calculation, our sample size was determined to be 376. Additionally, we accounted for a 20% attrition and non-response rate. Therefore, the total calculated sample size was around 450. However, in the actual study, a total of 500 participants were included. We invited HAs to participate in the study and wanted to provide an opportunity for all those interested in participating to do so. The study was conducted using an online questionnaire so the logistic requirements for additional participants were minimal. The survey was kept open for a longer time to enable HAs living in more remote areas to participate.

The inclusion criteria for this study were health assistants registered with the Nepal Health Professional Council, who had practiced for over two years, willing to participate, and providing informed consent. The study excluded participants who had practiced for less than 2 years,

participants who had changed their profession after obtaining the licensure, and participants who were not willing to participate.

We conducted a pilot test to assess the questionnaire's clarity, comprehension, and overall suitability for our target population. The pilot study involved 51 participants. The participants' feedback was valuable in refining and improving the questionnaire. To evaluate the questionnaire's reliability, we measured the Cronbach's alpha coefficient. The obtained value was 0.683, indicating satisfactory internal consistency among the questionnaire items. The Cronbach's alpha values if a particular statement was deleted is shown in the Appendix. The data from the pilot study was not included in the results.

Considering the feedback received from the pilot study and the reliability assessment, ambiguous or confusing questions were removed or rephrased, and response options were refined to enhance clarity and ease of understanding. The changes are highlighted in yellow in the questionnaire used during the pilot study in the Appendix. Data was coded and analyzed using SPSS software version 28, and descriptive statistics were used to summarize demographic information. The total knowledge score was calculated by adding the scores of statements in section three of the questionnaire. The statements with response options "Agree," "Disagree," and "Don't know" were coded based on their positive or negative nature. For positive statements, the responses "Disagree" and "Don't know," were coded one, while "Agree" was coded as two. Conversely, for negative statements, the coding was reversed, with "Agree" and "Don't know" assigned a value of one, and "Disagree" coded as two. One-Sample Kolmogorov-Smirnov Test was used to test the normality of the distribution of the total scores, and it revealed that the distribution was not normal. Knowledge levels were compared among different groups of respondents using appropriate statistical tests ($p < 0.05$). Correlation was conducted to examine the relationship between different variables.

The recruitment process for participants in this study used multiple channels targeting HAs. Initial outreach was done through a notice shared on the first author's personal social media page, leading to HAs expressing interest. These interested individuals were subsequently provided with an online Google Form for participation. Moreover, engagement was extended through a closed Facebook group moderated by one of the authors, with a membership of over 2300 HAs. Additionally, the study was also advertised during formal lectures delivered by the research team specifically tailored for HAs. This multi-faceted approach allowed for diverse participation and maximized the potential representation of HAs from across the country.

Data was obtained via an online Google Form. Personal identifying information was not collected. At the survey's inception, online informed consent was obtained from all participants. The participant information sheet and informed consent form were shared online, and these were previously approved by the ethical review board of the Nepal Health Research Council. The name, phone number and email address of the first author were included and respondents could contact him in case of any doubts or if they required more information. Only after the participants clicked the radio button indicating their willingness to participate did the survey open in a new window. The repository of collated data was exclusively accessible to the research team. The data was stored in an online password protected folder and was linked to the personal email accounts of the two authors. The research study (Proposal ID: 471–2022) was approved by the Nepal Health Research Council (NHRC) on December 1, 2022.

## Results

### Characteristics of study participants

There were a total of 500 participants for the main study. The majority were male (n = 391, 78.2%), followed by females (n = 108, 21.6%), and one participant selected 'Other' as their

gender identity. Most participants were between 25–29 years of age (n = 274, 54.8%), followed by those between 30–34 years (36.8%), while only a small percentage of participants were under 20 years (0.2%) or over 35 years of age (6.8%). Participants were from all seven provinces. Madhesh Province had the highest number of participants (159 participants, 31.8%), followed by Sudurpashchim Province (122 participants, 24.4%), while the other provinces had fewer participants. Rautahat had the highest number of participants among districts, followed by Sarlahi and Mahottari.

Most participants (43.7%) reported a monthly income above 30000 NPR (The exchange rate between the United States Dollar (USD) and the Nepalese Rupee (NPR) was approximately 1 USD to 132.04 NPR during the period of data collection), while only 26.1% of respondents earned below 10000 NPR a month. Most respondents (88.4%) had less than 5 years of work experience, while a smaller percentage (11.2%) had 5–10 years of work experience. Most respondents (47.8%) worked in government hospitals, while a significant proportion (31.6%) mentioned they were not working. A smaller proportion of the respondents were either working in private hospitals/clinics (18.4%). The vast majority of those who worked in the government sector did so in primary healthcare centers and health posts. The characteristics of the study participants are presented in Table 1.

Out of 500 HAs, 497 (99.4%) had heard about CPR, while 3 (0.6%) had not. Only 64 (12.8%) HAs had received training in CPR, while 436 (87.2%) had not. None of the HAs surveyed had ever performed CPR. Among them, 389 (77.8%) had attended theory classes on CPR during their three years of training, while 111 (22.2%) had not done so. Only 252 (50.4%) HAs had received practice sessions on CPR during their HA training (Table 2).

## Total scores among different subgroups

There were no significant differences in the total knowledge scores of HAs across different age groups (p-value = 0.338), but there were significant differences in the total knowledge scores of HAs across different provinces (p-value = 0.014) and gender (p-value = 0.001).

The knowledge scores for CPR varied among provinces, with Madhesh, Lumbini, Karnali, and Sudurpashchim provinces having higher median score (26.00) compared to Koshi, Bagmati, and Gandaki provinces (25.00). There was no significant difference in median total scores between participants who earned different salaries (p-value = 0.360) and participants with varying levels of work experience (p-value = 0.832).

## Responses for correct depth of CPR in adult, child, and infant

Participants were asked about the correct depth of CPR compression for adults, children, and infants, and their responses were recorded as correct, incorrect, or not answered. In the Adult category, only 36.8% of respondents answered correctly. The percentage was lower for child and infant at 21% and 17.4%.

## Knowledge score and factors related to CPR knowledge and training

There were significant differences in the distribution of total scores across most of the variables examined, except for the duration of practice of CPR. The variables examined include CPR knowledge, CPR training, theory of CPR, duration of CPR, practice of CPR, duration of practice, use of the defibrillator, and having heard about automated external defibrillator (AED).

Overall, the results suggest that variables related to CPR knowledge and training, such as whether someone has received CPR training or has knowledge of CPR, as well as variables related to performance, such as the depth of chest compressions and the use of a defibrillator, are important predictors of performance during CPR.

**Table 1. Demographic characteristics of participants and median and interquartile range of total knowledge scores among different groups of respondents.**

| Topic | Number | Percentage | Median Score | Interquartile Range (IQR) | P value |
|---|---|---|---|---|---|
| Participants | 500 | 100 | | | |
| Gender | | | | | |
| • Male | 391 | 78.2 | 26.00 | 2 | **0.001** |
| • Female | 108 | 21.6 | 25.00 | 3 | |
| • Other | 1 | 0.2 | 25.00 | 0 | |
| Age (in years) | | | | | |
| • 20–24 | 7 | 1.4 | 26.00 | 2 | 0.338 |
| • 25–29 | 274 | 54.8 | 26.00 | 3 | |
| • 30–34 | 184 | 36.8 | 25.00 | 3 | |
| • 35+ | 34 | 6.8 | 25 | 3 | |
| Province | | | | | |
| • Koshi Province | 36 | 7.2 | 25.00 | 3 | **0.014** |
| • Madhesh Province | 159 | 31.8 | 26.00 | 2 | |
| • Bagmati Province | 26 | 5.2 | 25.00 | 5 | |
| • Gandaki Province | 21 | 4.2 | 25.00 | 3 | |
| • Lumbini Province | 49 | 9.8 | 26.00 | 3 | |
| • Karnali Province | 87 | 17.4 | 26.00 | 3 | |
| • Sudurpashchim Province | 122 | 24.4 | 26.00 | 3 | |
| District | | | | | |
| • Rautahat | 33 | 6.6 | 27 | 3 | 0.248 |
| • Sarlahi | 27 | 5.4 | 27 | 2 | |
| • Mahottari | 22 | 4.4 | 25 | 3 | |
| • Others | 418 | 83.6 | 25 | 5 | |
| Income (Nepalese rupees) | | | | | |
| • Below 10000 | 130 | 26.1 | 26.00 | 3 | 0.360 |
| • 10000–20000 | 66 | 13.2 | 26.00 | 2 | |
| • 20000–30000 | 85 | 17.0 | 26.00 | 2 | |
| • Above 30000 | 218 | 43.7 | 26.00 | 3 | |
| • Missing | 1 | 0.2 | N/A | N/A | |
| Work Experience | | | | | |
| • < 5 years | 442 | 88.4 | 26.00 | 2 | 0.832 |
| • 5–10 years | 56 | 11.2 | 26.00 | 3 | |
| • > 10 years | 2 | 0.4 | 25.50 | N/A | |
| Employment | | | | | |
| • Government Hospitals | 239 | 47.8 | 26.00 | 3 | 0.062 |
| • Private Hospitals/Clinics | 92 | 18.4 | 26.00 | 3 | |
| • Not Working | 158 | 31.6 | 26.00 | 3 | |
| • Semi-Government Hospitals | 11 | 2.2 | 25.00 | 3 | |

N/A: Not available

This table displays the results of the Independent-Samples Kruskal-Wallis Test conducted to compare the differences in total knowledge scores among different groups of respondents. The p-value indicates the level of significance for the observed differences.

For the question "Have you heard about CPR (Cardiopulmonary Resuscitation)?" the median total knowledge score for those who answered "yes" to this question was 26, while the score for those who answered "no" was 19. The difference was not statistically significant. For the question "Have you received CPR training?" the median knowledge score for those who

**Table 2. Response frequency and percentage for questions on CPR training and theory.**

| Question | Response | Frequency | Percentage |
|---|---|---|---|
| 1. Have you heard about CPR? | Yes | 497 | 99.4% |
| | No | 3 | 0.6% |
| 2. Have you taken any training in CPR? | Yes | 64 | 12.8% |
| | No | 436 | 87.2% |
| 3. Have you ever performed CPR? | Yes | 0 | 0% |
| | No | 500 | 100% |
| 4. Did you attend any theory classes on CPR during your Health assistant training? | Yes | 389 | 77.8% |
| | No | 111 | 22.2% |
| 5. Did you receive any practice sessions on CPR during your Health assistant training? | Yes | 252 | 50.4% |
| | No | 248 | 49.6% |

answered yes to the question was 27, while the median score for those who answered no was 26 (p<0.001).

For the question "Did you attend any theory classes on CPR (Cardiopulmonary Resuscitation) during your 3 years of Health assistant training?" the median knowledge for those who answered "yes" was 26, while the median score for those who answered "no" was 25 (p = 0.005). Table 3 presents the median and interquartile range (IQR) values for different individual statements related to CPR knowledge and training.

## Correlation of CPR skills and work experience, training, and response to depth of CPR

Work experience has a significant positive correlation with correctly answering the depth of adult CPR (r = 0.137, p < 0.01) and depth in child CPR (r = 0.164, p < 0.01), which means that individuals with more work experience in CPR tend to perform better in these two aspects.

**Table 3. Median and IQR values for different individual statements.**

| Q. No | Question | Median | IQR | Most Responders Agree/Disagree |
|---|---|---|---|---|
| 1 | Cardiopulmonary Resuscitation is an emergency procedure to save life | 2.00 | 0.00 | Agree |
| 2 | CPR is only done for myocardial infarction | 2.00 | 0.00 | Disagree |
| 3 | CPR is done for cardiac arrest | 2.00 | 0.00 | Agree |
| 4 | CPR done 10 minutes after cardiac arrest has low survival chance | 2.00 | 1.00 | Agree |
| 5 | The compression ratio in 2 rescuer adult CPR is 30:2 | 2.00 | 0.00 | Agree |
| 6 | CPR is only done in hospital setting | 1.00 | 0.00 | Disagree |
| 7 | The American Heart Association (AHA) follows the acronym A-B-C during the CPR A: Airway B: Breathing C: Chest compression | 2.00 | 0.00 | Agree |
| 8 | We should tap the hand to check the person's responsiveness | 1.00 | 0.00 | Disagree |
| 9 | The compression is done with the heel of the dominant hand | 2.00 | 1.00 | Agree |
| 10 | For quality CPR, 5 cycles should be performed in 2 minutes | 2.00 | 1.00 | Agree |
| 11 | We should not allow the chest to recoil during compression | 2.00 | 0.00 | Disagree |
| 12 | If there is pulse but no breathing, do not continue compression | 2.00 | 1.00 | Agree |
| 13 | The depth of the quality CPR in adult is at least 2 inch (5 cm) | 2.00 | 0.00 | Agree |
| 14 | The rate of the quality CPR in adult is 100–120 compression per minute | 2.00 | 1.00 | Agree |
| 15 | The first step in AHA chain of survival for adult in In-Hospital Cardiac Arrest (IHCA) is high quality CPR. | 2.00 | 0.00 | Agree |
| 16 | Adult cardiac arrest algorithm recommends 1 breath every 6 seconds under advanced airway | 2.00 | 1.00 | Agree |

**Table 4. Significant correlations between work experience, CPR knowledge and training, and CPR performance measures.**

| Comparison | Pearson Correlation | Sig. (2-tailed) | Conclusion |
|---|---|---|---|
| Work experience vs. Depth (adult) response | 0.137 | **0.002** | Those with more work experience had better knowledge of the depth of compression in adults |
| Work experience vs. Depth (child) response | 0.164 | **<0.001** | Those with more work experience had better knowledge of the depth of compression in children. |

** Correlation is significant at the 0.05 level (2-tailed).

Table 4 provides a summary of significant correlations between work experience, CPR knowledge, and CPR performance measures.

## Discussion

The findings revealed that most participants had heard about CPR, indicating a basic awareness of the procedure. However, only a small proportion of HAs had received formal training in CPR, suggesting a lack of practical skills. None of the participants had performed CPR, further emphasizing the need for more practical training opportunities.

The study also highlighted the disparity between theory and practice in CPR training. While most HAs had attended theory classes on CPR during their training, few had received practical training or practice sessions. This finding raises concerns about the effectiveness of the current training curriculum and the preparedness of HAs to perform CPR in real-life situations.

Our study aligns with similar studies conducted in different settings. An observational study assessed the knowledge and attitudes of medical and paramedical professionals, including doctors, nurses, and HAs. The study, which involved 121 participants, concluded that health personnel, including HAs lacked sufficient knowledge of CPR and basic life support (BLS) [14]. Similarly, another study focused on the competency of BLS among 95 healthcare workers, including PCL in Nursing (PCL Nursing), HAs, Auxiliary Nurse Midwives (ANM), and Community Medicine Assistants (CMA) [15]. The findings indicate that most participants demonstrated an insufficient understanding of BLS techniques. These two studies provide valuable context and highlight the need to assess and enhance the CPR knowledge of this healthcare professional group. While there have been no licensure examinations for HAs, the NHPC released a syllabus in 2021, initiating licensure exams for them [10]. Five hours of theory and 4 hours of tutorial are allocated for CPR [9].

Another study among allied health university students in Jordan revealed poor CPR knowledge, emphasizing the need for targeted awareness programs and mandatory training courses [16]. Similarly, among Iranian medical interns, insufficient knowledge and practical skills for performing CPR were found due to a lack of training opportunities and dedicated emergency medicine wards [17]. Doctors at an academic hospital in South Africa also exhibited inadequate CPR knowledge, primarily attributed to a lack of regular training opportunities [18]. In a study conducted among podiatrists in New Zealand, high levels of CPR confidence were reported, but gaps in CPR knowledge were identified despite having compulsory CPR certification [19]. Furthermore, a study conducted in Tanzania among healthcare providers from various departments highlighted poor CPR knowledge and skills, despite self-reported CPR experience [20].

A national report showed variations in the human development indices (HDI) across provinces [21]. The median CPR knowledge score (26) was higher in Karnali Province, which

ranks sixth in HDI. Lumbini Province, ranking fourth in HDI, also had a median CPR knowledge score of 26. The scores were lower among Koshi, Bagmati and Gandaki provinces that higher HDI. This suggests that CPR knowledge levels may not directly correlate with the overall human development rankings.

The HAs course is divided into three years. The first year is a non-clinical year where students cover subjects such as English, Nepali, social studies, anatomy and physiology, botany, zoology, chemistry, physics, mathematics, and computer science. The second year and the third year comprise the clinical subjects along with first aid, primary health care/ family health, health education, epidemiology, community diagnosis, environmental health, and health management. Both the second and third years consist of 60% theory and 40% practical. The second half of the third year is exclusively dedicated to comprehensive clinical practice and comprehensive field practice. The medium of instruction is English and/or Nepali [9].

Our findings reinforce the importance of enhancing CPR training programs to bridge the gap between knowledge and practical application of CPR. By addressing training insufficiencies and providing regular updates, healthcare professionals can improve their competency in performing life-saving CPR interventions.

To address these gaps, it is essential to incorporate practical training and regular refresher courses on CPR into the curriculum for HAs. Practical application and hands-on practice are crucial for developing and maintaining the skills and confidence needed for effective CPR. By including practical training in the curriculum, HAs can gain the necessary skills to provide life-saving interventions in emergency situations. Additionally, the study's findings have implications for curriculum development, updates, and policymaking in healthcare delivery. The results can guide the integration of life-saving training into the national curriculum for HAs and the NHPC health assistant's licensure exam syllabus. By addressing the gaps in CPR knowledge and skills, the quality of healthcare delivery can be enhanced, particularly in remote and rural areas.

It is important to acknowledge the limitations of this study. Our study specifically targeted HAs, and thus we cannot generalize the findings to encompass other health professionals. This study did not include a practical assessment of CPR skills, which could provide a more comprehensive understanding of HAs' competency in performing CPR. There was a potential for selection bias due to the convenience sampling methods adopted in our study. Although social media is accessible across a wide geographical area of the country, there is still a possibility that the online survey may have excluded HAs who are unfamiliar with online platforms, resulting in a biased sample. The distribution of HAs across provinces was not compared to that of the study population. Further studies are required.

In conclusion, the study highlighted the need for improved CPR training among HAs in Nepal. Practical training, along with theoretical knowledge, is crucial for ensuring the effective delivery of life-saving interventions. The findings underscore the importance of integrating practical training into the curriculum and providing regular refresher courses to enhance the knowledge and skills of HAs. Addressing these gaps can strengthen healthcare delivery in Nepal, particularly in remote and rural regions with limited access to physicians. The results of this study can serve as a base for future research, curriculum development, and policymaking in healthcare.

## Supporting information

**S1 File. Cronbach's alpha pilot study.**
(DOC)

**S2 File. Pilot study questionnaire.**
(DOCX)

## Acknowledgments

We gratefully acknowledge the invaluable contributions of Mukesh Adhikari, Nour Ammar, Om Murti Anil, and Pratik Khanal in meticulously validating the content of our research questionnaire.

## Author Contributions

**Conceptualization:** Bivek Singh, Pathiyil Ravi Shankar.

**Data curation:** Bivek Singh, Pathiyil Ravi Shankar.

**Formal analysis:** Pathiyil Ravi Shankar.

**Investigation:** Bivek Singh.

**Methodology:** Bivek Singh, Pathiyil Ravi Shankar.

**Project administration:** Bivek Singh.

**Supervision:** Pathiyil Ravi Shankar.

**Validation:** Bivek Singh, Pathiyil Ravi Shankar.

**Writing – original draft:** Bivek Singh, Pathiyil Ravi Shankar.

**Writing – review & editing:** Pathiyil Ravi Shankar.

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
