## [Decision Letter · Decision Letter 0]

8 Aug 2023

PONE-D-23-18050Knowledge regarding cardiopulmonary resuscitation among health assistants in Nepal: A cross-sectional studyPLOS ONE

Dear Dr. SINGH,

Thank you for submitting your manuscript to PLOS ONE. After careful consideration, we feel that it has merit but does not fully meet PLOS ONE’s publication criteria as it currently stands. Therefore, we invite you to submit a revised version of the manuscript that addresses the points raised during the review process.

I personally read the manuscript and go through comments by reviewers. Based on my evaluation, the manuscript subject is of interest and with substantial revision with re-analysis and write up it can be reconsidered for the further processing. Please carefully address the comments put forth by reviewers with appropriate rebuttal.  

We look forward to receiving your revised manuscript.

Kind regards,

Dhan Bahadur Shrestha, MD

Academic Editor

PLOS ONE

“I have read the journal's policy and the authors of this manuscript have the following competing interests.

[P Ravi Shankar is an academic editor of PLOS ONE]”

Please respond by return email with your amended Competing Interests Statement and we will change the online submission form on your behalf.

Reviewers' comments:

Reviewer's Responses to Questions

**Comments to the Author**

1. Is the manuscript technically sound, and do the data support the conclusions?

Reviewer #1: Yes

Reviewer #2: No

Reviewer #3: Yes

2. Has the statistical analysis been performed appropriately and rigorously? 

Reviewer #1: No

Reviewer #2: Yes

Reviewer #3: Yes

3. Have the authors made all data underlying the findings in their manuscript fully available?

Reviewer #1: No

Reviewer #2: Yes

Reviewer #3: Yes

4. Is the manuscript presented in an intelligible fashion and written in standard English?

Reviewer #1: Yes

Reviewer #2: No

Reviewer #3: Yes

5. Review Comments to the Author

Reviewer #1: The manuscript needs major revision on following aspects:

TITLE

Please write in journal’s order: first full title then short title

ABSTRACT

Please include information related to knowledge scores and other relevant aspects in results section because this is one of the main findings of your study

INTRODUCTION

This section is very lengthy and redundant. It includes lots of unnecessary information which can be included in DISCUSSION section (second and third paragraph) later.

Please summarize/state the main aim/objective of your study rather than splitting it into general and specific objectives.

(Please follow the PLOS ONE submission guidelines on what to include in this section: https://journals.plos.org/plosone/s/submission-guidelines#loc-title )

MATERIALS AND METHODS

Please mention which online platform/media you used for the survey. And, clarify how confidentiality and privacy was maintained

Please cite the reference for 2020 AHA guidelines

Please spell the single digit (eg. one for 1)

RESULTS

This section needs significant revisions as follows:

There is no need to repeat information which has been already elaborated in MATERIALS AND METHODS section (first and an initial portion of second paragraph)

In Table 1:

No need to repeat % once you have mentioned it in heading list

Please clearly write ‘median score’ rather than median only so that readers would easily understand

Please clarify what N/A means in legends of table

It would be better if significant p-values are made bold

Please mention in table legend which statistical test you applied to compare differences and find p-values

In METHODS section, you have stated that the data was non-normal, and therefore you have calculated median and IQR. To the contrary, in the text you have mentioned mean values. Please maintain uniformity throughout the manuscript. Use only median/IQR to describe in the text.

In subsection “TOTAL SCORES AMONGN DIFFERENT GROUPS (224-229)”

The interpretation does not look correct. Koshi, Gandaki and Bagmati provinces have average scores of 25.0 and others have 26.0. You need to explain them accordingly.

Please omit this sentence “Koshi province exhibiting higher CPR scores”

In Table 4:

Could you please clarify why significance level for correlation coefficient was taken at 0.01 rather than 0.05 that you had taken in table 1. Ideally, the same level of significance should be taken throughout the analysis for same study sample.

DISCUSSION

Fourth paragraph (292-294): I think you need to rewrite this information. This is not in line with your study findings “Highest score in Karnali and lowest score in lumbini”

REFERENCES

PLOS uses the reference style outlined by the International Committee of Medical Journal Editors (ICMJE), also referred to as the “Vancouver” style. Please make sure that all the references are in line with this.

(https://journals.plos.org/plosone/s/submission-guidelines#loc-references )

Reviewer #2: After going through the study entitled "Knowledge regarding cardiopulmonary resuscitation among health assistants in Nepal: A cross-sectional study", I have the following comments to make.

Research hypothesis:

The manuscript lacks a clear research question or hypothesis. While the aim is to evaluate CPR knowledge among HAs, it would be beneficial to explicitly state the specific research question or hypothesis being tested.

Introduction:

The introduction section provides an overview of the healthcare system in Nepal and the role of HAs, but it would benefit from more comprehensive background information and relevant references. Providing context about the healthcare challenges and the importance of CPR knowledge would strengthen the introduction.

Methods: I have a major concern regarding the methodology of the study. The methodology should be transparent and reproducible.

1. Sample Size Calculation and Participant Recruitment:

I share a concern regarding the sample size calculation. The authors state that the total population of registered health assistants in Nepal is 17,363, and they aimed to have a sample size of 376 based on a 50% distribution, a 95% confidence interval, and a 5% margin of error. However, they ultimately recruited 500 participants for the study, which is significantly larger than the calculated sample size. The authors should provide a clear justification for deviating from the initial sample size calculation. Additionally, they need to explain how they recruited participants from different regions of Nepal, as this information is crucial for assessing the generalizability of their findings.

2. Questionnaire Distribution and Informed Consent:

The methodology does not clearly describe how the questionnaire was distributed among the participants. Since the study employed an online survey, it is important to understand how participants were selected and how the questionnaire was disseminated. In the limitations section, the authors have mentioned social media was used, however, it has not been specified in the methods section. Furthermore, the authors need to provide a detailed explanation of how written informed consent was obtained from the participants. Given that the study was conducted online, it is unclear how the authors ensured the participants' understanding and voluntary agreement to participate.

3. Validity of the Questionnaire and Pretesting:

I have concerns regarding the validity of the questionnaire and the pretesting process. The authors briefly mention conducting a pretest with 51 participants, which accounted for approximately 10% of the study sample. However, it is unclear whether these participants were included in the final analysis or if they were distinct from the study participants. [Although the authors state these were excluded, line 159, and that is the current practice, the results show 500 samples]. Additionally, the methodology lacks information on how the feedback received from the pilot study was used to refine and improve the questionnaire. A clear description of the modifications made to the questionnaire based on the pilot study feedback is necessary to establish the questionnaire's validity. Please mention the parts of the questionnaire and the total number of the questions. As the authors mention Cronbach’s alpha value was 0.7, it would had been better if the questionnaire and the calculation were provided in the supplementary file so that the same validated questionnaire could be used in future studies. Furthermore, a mention of non-response rate, incompletely filled forms, non consenting participants would add validity and rigor of data collection process.

Results:

The results section provides a summary of the characteristics of the study participants. The sample size mentioned in the results (n=500) does not match the total sample size mentioned in the methods section (approximately 450). The authors need to address this discrepancy and provide a clear explanation. It seems the participants from the pilot study were included in the main study, it questions the overall validity of the study findings (450+50=500)???

Discussion:

The discussion section of the manuscript provides a comprehensive analysis and interpretation of the study's findings. It discusses the knowledge gaps and training insufficiencies in CPR among HAs in Nepal, drawing comparisons to similar studies conducted in other settings. The discussion also highlights the importance of practical training and hands-on practice in bridging the gap between CPR knowledge and its effective application.

While the discussion section addresses the implications of the findings, it could have delved deeper into the specific recommendations for improving CPR training and curriculum for HAs. Providing more specific suggestions would enhance the practical relevance of the study.

The discussion section does not explicitly mention the significance of the study's results. It would be helpful to explicitly state how the study contributes to the existing knowledge and fills the research gap.

Language: A thorough proof-reading by English language expert or native speaker would enhance the overall language of the manuscript.

Reference: The referencing style used in the manuscript is not uniform. I suggest the authors use referencing software available so that the citation and referencing errors are minimized.

Overall, while the study's objective and design are appropriate, there are several methodological concerns that need to be addressed. The authors should provide a justification for the larger sample size, clarify participant recruitment methods, explain the process of questionnaire distribution and informed consent, and provide more details regarding the validity of the questionnaire and the modifications made based on the pretest. Addressing these concerns will enhance the study's validity and improve the overall quality of the research.

Reviewer #3: The manuscript is well written. It is an essential topic, especially in South Asia. Although some grammatical corrections need to be made, the overall manuscript seems fine. The study highlights the need of more practical training in the remote areas. The statistical analyses are also avidly simple and easier to understand and comprehend. I hope we can consider it for publication after minor editing.

6. PLOS authors have the option to publish the peer review history of their article (what does this mean?). If published, this will include your full peer review and any attached files.

Reviewer #1: No

Reviewer #2: **Yes: **Alok Atreya

Reviewer #3: No

---

## [Author Response · Author response to Decision Letter 0]

11 Sep 2023

8th September 2023

Editorial Office, PLOS ONE

Dear Editor and Reviewers,

We would like to express our sincere gratitude for the time and effort you have dedicated to reviewing our manuscript titled "Knowledge Regarding Cardiopulmonary Resuscitation Among Health Assistants in Nepal: A Cross-Sectional Study," submission ID PONE-D-23-18050. We appreciate your valuable feedback, which has undoubtedly contributed to enhancing the quality and rigor of the manuscript.

In response to the comments and suggestions provided by the reviewers, we have made the following revisions and clarifications:

Reviewer 1:

Title: Please write in journal’s order: first full title then short title

We have reordered the titles as per the journal's requirements: "Knowledge Regarding Cardiopulmonary Resuscitation Among Health Assistants in Nepal: A Cross-Sectional Study" (Full Title), and "CPR Knowledge Among Health Assistants in Nepal: A Cross-Sectional Study" (Short Title).

Abstract: Please include information related to knowledge scores and other relevant aspects in results section because this is one of the main findings of your study

We have included additional information related to knowledge scores in the Results section of the abstract to provide a more comprehensive overview of the study's findings (lines 42 to 45).

Introduction: This section is very lengthy and redundant. It includes lots of unnecessary information which can be included in DISCUSSION section (second and third paragraph) later.

The Introduction section has been revised to be more concise and focused. It now provides essential background information while avoiding unnecessary redundancy.

Please summarize/state the main aim/objective of your study rather than splitting it into general and specific objectives.

We have summarized the main aim/objective of the study, which is to evaluate CPR knowledge among Health Assistants in Nepal and explore differences among subgroups of respondents (lines 116 to 118).

Materials and Methods: � Please mention which online platform/media you used for the survey. And, clarify how confidentiality and privacy was maintained

We have provided information about the online platform/media used for the survey and clarified how confidentiality and privacy were maintained. (lines 214 to 232).

Please cite the reference for 2020 AHA guidelines

The reference to the 2020 AHA guidelines has been cited in the manuscript (line 152).

Please spell the single digit (eg. one for 1)

Single digits have been spelled out as per your suggestion.

Results:

This section needs significant revisions as follows:

There is no need to repeat information which has been already elaborated in MATERIALS AND METHODS section (first and an initial portion of second paragraph)

The repetition in the Results section, which was also present in the Materials and Methods section, has been removed as per your recommendation.

In Table 1:

No need to repeat % once you have mentioned it in heading list

In Table 1, I have removed the repetition of '%' and clarified the 'median score' for better understanding.

Please clearly write ‘median score’ rather than median only so that readers would easily understand

Done (lines 256 to 263)

Please clarify what N/A means in legends of table

'N/A' in the table legend has been explained as 'Not available.'

It would be better if significant p-values are made bold 

Significant p-values have been made bold in the table, and the statistical test used has been mentioned.

Please mention in table legend which statistical test you applied to compare differences and find p-values 

Done (lines 261 to 263). 

In METHODS section, you have stated that the data was non-normal, and therefore you have calculated median and IQR. To the contrary, in the text you have mentioned mean values. Please maintain uniformity throughout the manuscript. Use only median/IQR to describe in the text.

Throughout the manuscript, we have maintained uniformity by using median/IQR to describe non-normal data.

In subsection “TOTAL SCORES AMONGN DIFFERENT GROUPS (224-229)”

The interpretation does not look correct. Koshi, Gandaki and Bagmati provinces have average scores of 25.0 and others have 26.0. You need to explain them accordingly.

The interpretation of median scores among different provinces has been clarified (lines 277 to 279).

Please omit this sentence “Koshi province exhibiting higher CPR scores”

The sentence mentioning "Koshi province exhibiting higher CPR scores" has been removed.

In Table 4: Could you please clarify why significance level for correlation coefficient was taken at 0.01 rather than 0.05 that you had taken in table 1. Ideally, the same level of significance should be taken throughout the analysis for same study sample.

The significance level for correlation coefficients has been set at 0.05, consistent with the analysis in Table 1 (line 334).

Discussion:

Fourth paragraph (292-294): I think you need to rewrite this information. This is not in line with your study findings “Highest score in Karnali and lowest score in lumbini”

The fourth paragraph has been rephrased to align with the study's findings.

Reviewer 2

Research hypothesis:

The manuscript lacks a clear research question or hypothesis. While the aim is to evaluate CPR knowledge among HAs, it would be beneficial to explicitly state the specific research question or hypothesis being tested.

The hypothesis has been added as suggested (lines 117 and 118). 

Introduction:

The introduction section provides an overview of the healthcare system in Nepal and the role of HAs, but it would benefit from more comprehensive background information and relevant references. Providing context about the healthcare challenges and the importance of CPR knowledge would strengthen the introduction.

We have tried to provide more information. However, the first reviewer has suggested a significant reduction in the length of the Introduction and we have tried to address conflicting requirements. The information has also been provided in the Discussion section.. 

Methods: I have a major concern regarding the methodology of the study. The methodology should be transparent and reproducible.

1. Sample Size Calculation and Participant Recruitment:

I share a concern regarding the sample size calculation. The authors state that the total population of registered health assistants in Nepal is 17,363, and they aimed to have a sample size of 376 based on a 50% distribution, a 95% confidence interval, and a 5% margin of error. However, they ultimately recruited 500 participants for the study, which is significantly larger than the calculated sample size. The authors should provide a clear justification for deviating from the initial sample size calculation. Additionally, they need to explain how they recruited participants from different regions of Nepal, as this information is crucial for assessing the generalizability of their findings.

The method of recruiting participants for the study has been mentioned in the Methods section. We kept the study open as long as there were requests from participants to enroll. We were also of the opinion that participants from the more remote districts and provinces may require more time to participate as they may face challenges in internet access (lines 166 to 171). The distribution of participants from different regions and provinces is shown in table 1. Getting participants from more remote areas required more promotion and interaction with the participants by the authors. 

Questionnaire Distribution and Informed Consent:

The methodology does not clearly describe how the questionnaire was distributed among the participants. Since the study employed an online survey, it is important to understand how participants were selected and how the questionnaire was disseminated. In the limitations section, the authors have mentioned social media was used, however, it has not been specified in the methods section. Furthermore, the authors need to provide a detailed explanation of how written informed consent was obtained from the participants. Given that the study was conducted online, it is unclear how the authors ensured the participants' understanding and voluntary agreement to participate.

This has been mentioned on page 9 (lines 214 to 222). 

Validity of the Questionnaire and Pretesting:

I have concerns regarding the validity of the questionnaire and the pretesting process. The authors briefly mention conducting a pretest with 51 participants, which accounted for approximately 10% of the study sample. However, it is unclear whether these participants were included in the final analysis or if they were distinct from the study participants. [Although the authors state these were excluded, line 159, and that is the current practice, the results show 500 samples].

The Methods and the Results section clarifies that there were 51 participants for the pilot testing and 500 participants for the main study. We have provided details about the sampling process in the Methods section as requested previously. 

Additionally, the methodology lacks information on how the feedback received from the pilot study was used to refine and improve the questionnaire. A clear description of the modifications made to the questionnaire based on the pilot study feedback is necessary to establish the questionnaire's validity. Please mention the parts of the questionnaire and the total number of the questions. As the authors mention Cronbach’s alpha value was 0.7, it would had been better if the questionnaire and the calculation were provided in the supplementary file so that the same validated questionnaire could be used in future studies. Furthermore, a mention of non-response rate, incompletely filled forms, non consenting participants would add validity and rigor of data collection process.

The Cronbach alpha values was 0.683. This is addressed in lines 186 to 193. The original questionnaire with statements removed highlighted in yellow is shown in the Appendix. Cronbach’s alpha values of the different questions and the values if each of the statements were deleted are shown in the Appendix. There were no major changes in values if a particular statement was deleted. One of the statements and another question were removed as suggested by experts during the content validation process. Those who consented to participate had completed the required demographic information, and the different statements and provided other information. Hence, they were included in the final analysis. We did not obtain information on those who did not agree to participate. 

Results:

The results section provides a summary of the characteristics of the study participants. The sample size mentioned in the results (n=500) does not match the total sample size mentioned in the methods section (approximately 450). The authors need to address this discrepancy and provide a clear explanation. It seems the participants from the pilot study were included in the main study, it questions the overall validity of the study findings (450+50=500)???

Thank you for your concern. As we have mentioned previously, participants from the pilot study were not included in the main study. We continued to keep data collection open till there were no further requests from the respondents and we wanted to provide participants from the more remote areas with opportunities to participate in the study. 

Specific recommendations and suggestions for improving CPR training and curriculum for Health Assistants have been added for practical relevance.

Discussion:

The discussion section of the manuscript provides a comprehensive analysis and interpretation of the study's findings. It discusses the knowledge gaps and training insufficiencies in CPR among HAs in Nepal, drawing comparisons to similar studies conducted in other settings. The discussion also highlights the importance of practical training and hands-on practice in bridging the gap between CPR knowledge and its effective application.

We thank the reviewer for the comments. 

While the discussion section addresses the implications of the findings, it could have delved deeper into the specific recommendations for improving CPR training and curriculum for HAs. Providing more specific suggestions would enhance the practical relevance of the study.

We provided recommendations to improve HA training on page 21 (lines 396 to 404). We have added to these recommendations. 

The discussion section does not explicitly mention the significance of the study's results. It would be helpful to explicitly state how the study contributes to the existing knowledge and fills the research gap.

The significance of the study's results and how they contribute to existing knowledge have been explicitly mentioned. There is a need for further studies to corroborate our results. 

Language: Language: A thorough proof-reading by English language expert or native speaker would enhance the overall language of the manuscript.

We have ensured thorough proofreading to enhance the manuscript's language quality.

Reference: Reference: The referencing style used in the manuscript is not uniform. I suggest the authors use the referencing software available so that the citation and referencing errors are minimized.

The referencing style has been standardized to the Vancouver style.

Overall, while the study's objective and design are appropriate, there are several methodological concerns that need to be addressed. The authors should provide a justification for the larger sample size, clarify participant recruitment methods, explain the process of questionnaire distribution and informed consent, and provide more details regarding the validity of the questionnaire and the modifications made based on the pretest. Addressing these concerns will enhance the study's validity and improve the overall quality of the research.

We have provided this information in the manuscript and have tried to address this in the revision letter. 

Reviewer #3: The manuscript is well written. It is an essential topic, especially in South Asia. Although some grammatical corrections need to be made, the overall manuscript seems fine. The study highlights the need of more practical training in the remote areas. The statistical analyses are also avidly simple and easier to understand and comprehend. I hope we can consider it for publication after minor editing.

We thank the reviewer for the comments. The manuscript has been copyedited for language and grammar. 

The revisions have been highlighted using red font in the revised manuscript. 

We kindly request that you consider the revised manuscript for publication in PLOS ONE, and we are open to further discussions or clarifications if needed. Thank you once again for your time and consideration.

Sincerely,

Bivek Singh

P Ravi Shankar

---

## [Decision Letter · Decision Letter 1]

3 Oct 2023

PONE-D-23-18050R1Knowledge regarding cardiopulmonary resuscitation among health assistants in Nepal: A cross-sectional studyPLOS ONE

Dear Dr. SINGH,

Thank you for submitting your manuscript to PLOS ONE. After careful consideration, we feel that it has merit but does not fully meet PLOS ONE’s publication criteria as it currently stands. Therefore, we invite you to submit a revised version of the manuscript that addresses the points raised during the review process.

 Myself review the manuscript and comments from reviewers. Majority of comments from reviewers in earlier phase of review well addressed however there is a comment by Reviewer 1 to address in introduction section and a comment regarding references need to be addressed.Also, overall referencing is not alligned to PLOS guideline "References are listed at the end of the manuscript and numbered in the order that they appear in the text. In the text, cite the reference number in square brackets (e.g., “We used the techniques developed by our colleagues [19] to analyze the data”). PLOS uses the numbered citation (citation-sequence) method and first six authors, et al." Please kindly revise throughly considering the PLOS authors guideline cheslist https://journals.plos.org/plosone/s/submission-guidelines.

We look forward to receiving your revised manuscript.

Kind regards,

Dhan Bahadur Shrestha, MD

Academic Editor

PLOS ONE

Journal Requirements:

Additional Editor Comments:Majority of comments from reviewers in earlier reviews well addressed however there is a comment by Reviewer 1 to address in introduction section and a comment regarding references need to be addressed.Also, overall referencing is not alligned to PLOS guideline "References are listed at the end of the manuscript and numbered in the order that they appear in the text. In the text, cite the reference number in square brackets (e.g., “We used the techniques developed by our colleagues [19] to analyze the data”). PLOS uses the numbered citation (citation-sequence) method and first six authors, et al." Please kindly revise throughly considering the PLOS authors guideline cheslist https://journals.plos.org/plosone/s/submission-guidelines.

Reviewers' comments:

Reviewer's Responses to Questions

**Comments to the Author**

1. If the authors have adequately addressed your comments raised in a previous round of review and you feel that this manuscript is now acceptable for publication, you may indicate that here to bypass the “Comments to the Author” section, enter your conflict of interest statement in the “Confidential to Editor” section, and submit your "Accept" recommendation.

Reviewer #1: All comments have been addressed

Reviewer #2: All comments have been addressed

2. Is the manuscript technically sound, and do the data support the conclusions?

Reviewer #1: Yes

Reviewer #2: Yes

3. Has the statistical analysis been performed appropriately and rigorously? 

Reviewer #1: Yes

Reviewer #2: I Don't Know

4. Have the authors made all data underlying the findings in their manuscript fully available?

Reviewer #1: Yes

Reviewer #2: Yes

5. Is the manuscript presented in an intelligible fashion and written in standard English?

Reviewer #1: Yes

Reviewer #2: Yes

6. Review Comments to the Author

Reviewer #1: 1. Please omit this sentence from the introduction section "The null hypothesis proposes that there is no significant

100 difference in knowledge scores according to respondents’ demographic and other characteristics."

2. Please make sure that reference 13 is correct in line with Vancouver style

Reviewer #2: Thank you for re-inviting me to review the manuscript. After going through the author's response to the reviewer comments on the revised manuscript submitted to PLOS ONE, the authors have done a thorough job of addressing the major concerns raised during the initial review process. Here are my thoughts on some key points:

Sample Size and Recruitment:

- The authors have provided more details on participant recruitment, including keeping the survey open to allow more participants from remote areas. They have justified the larger final sample size.

Questionnaire Distribution and Consent:

- The process of online questionnaire distribution via social media and obtaining informed consent is now clearly described.

Questionnaire Validity and Pretesting:

- The authors have clarified that pilot study participants were excluded from the final analysis. Details are provided on content validation and questionnaire refinement based on expert input. The supplementary material with the questionnaire and Cronbach's alpha calculations enhances transparency.

Results:

- The sample size discrepancy between methods and results has been addressed. The authors clearly state pilot study participants were not included in the final analysis.

Discussion:

- Specific, practical recommendations for improving CPR training for health assistants have now been included, enhancing the manuscript's relevance.

Overall, the authors have been responsive to the major critiques and suggestions. The revised manuscript reads much stronger in terms of methodology and transparency. The additional details provided instill greater confidence in the validity of the study and its conclusions. I would recommend accepting this revised manuscript for publication in PLOS ONE, pending any final minor edits during the production process. The authors have undertaken considerable effort to improve the manuscript quality based on the reviewer feedback.

7. PLOS authors have the option to publish the peer review history of their article (what does this mean?). If published, this will include your full peer review and any attached files.

Reviewer #1: No

Reviewer #2: **Yes: **Alok Atreya

---

## [Author Response · Author response to Decision Letter 1]

4 Oct 2023

4th October 2023

To

The Editor

PLoS One

Sub: Submission of the revised version of manuscript PONE-D-23-18050R

Dear Editor

We are submitting the revised version of the manuscript PONE-D-23-18050R. The manuscript has been revised and the references have been reformatted.

The response to specific comments follows.

Please review your reference list to ensure that it is complete and correct. If you have cited papers that have been retracted, please include the rationale for doing so in the manuscript text or remove these references and replace them with relevant current references. Any changes to the reference list should be mentioned in the rebuttal letter that accompanies your revised manuscript. If you need to cite a retracted article, indicate the article’s retracted status in the References list and also include a citation and full reference for the retraction notice.

We have edited the references as suggested. 

Additional Editor Comments:

Majority of comments from reviewers in earlier reviews well addressed however there is a comment by Reviewer 1 to address in introduction section and a comment regarding references need to be addressed.

Also, overall referencing is not alligned to PLOS guideline "References are listed at the end of the manuscript and numbered in the order that they appear in the text. In the text, cite the reference number in square brackets (e.g., “We used the techniques developed by our colleagues [19] to analyze the data”). PLOS uses the numbered citation (citation-sequence) method and first six authors, et al." Please kindly revise throughly considering the PLOS authors guideline cheslist https://journals.plos.org/plosone/s/submission-guidelines.

We have revised the reference formatting and citation in the text. 

Reviewers' comments:

Reviewer #1: 1. Please omit this sentence from the introduction section "The null hypothesis proposes that there is no significant

100 difference in knowledge scores according to respondents’ demographic and other characteristics."

The sentence has been omitted. 

2. Please make sure that reference 13 is correct in line with Vancouver style

The reference has been edited. 

Reviewer #2: Thank you for re-inviting me to review the manuscript. After going through the author's response to the reviewer comments on the revised manuscript submitted to PLOS ONE, the authors have done a thorough job of addressing the major concerns raised during the initial review process. Here are my thoughts on some key points:

Sample Size and Recruitment:

- The authors have provided more details on participant recruitment, including keeping the survey open to allow more participants from remote areas. They have justified the larger final sample size.

Thank you.

Questionnaire Distribution and Consent:

- The process of online questionnaire distribution via social media and obtaining informed consent is now clearly described.

Thank you.

Questionnaire Validity and Pretesting:

- The authors have clarified that pilot study participants were excluded from the final analysis. Details are provided on content validation and questionnaire refinement based on expert input. The supplementary material with the questionnaire and Cronbach's alpha calculations enhances transparency.

Thank you.

Results:

- The sample size discrepancy between methods and results has been addressed. The authors clearly state pilot study participants were not included in the final analysis.

Thank you.

Discussion:

- Specific, practical recommendations for improving CPR training for health assistants have now been included, enhancing the manuscript's relevance.

Thank you.

Overall, the authors have been responsive to the major critiques and suggestions. The revised manuscript reads much stronger in terms of methodology and transparency. The additional details provided instill greater confidence in the validity of the study and its conclusions. I would recommend accepting this revised manuscript for publication in PLOS ONE, pending any final minor edits during the production process. The authors have undertaken considerable effort to improve the manuscript quality based on the reviewer feedback.

Thank you.

The changes have been carried out using track changes. 

Hoping for a favorable consideration 

With regards

Dr Singh

Dr Shankar

---

## [Editor Report · Decision Letter 2]

11 Oct 2023

Knowledge regarding cardiopulmonary resuscitation among health assistants in Nepal: A cross-sectional study

PONE-D-23-18050R2

Dear Dr. SINGH,

We’re pleased to inform you that your manuscript has been judged scientifically suitable for publication and will be formally accepted for publication once it meets all outstanding technical requirements.

Kind regards,

Dhan Bahadur Shrestha, MBBS

Academic Editor

PLOS ONE

---

## [Editor Report · Acceptance letter]

16 Oct 2023

PONE-D-23-18050R2 

"Knowledge Regarding Cardiopulmonary Resuscitation Among Health Assistants in Nepal: A Cross-Sectional Study" 

Dear Dr. Singh:

I'm pleased to inform you that your manuscript has been deemed suitable for publication in PLOS ONE. Congratulations! Your manuscript is now with our production department. 

Kind regards, 

on behalf of

Dr. Dhan Bahadur Shrestha 

Academic Editor

PLOS ONE